# Data-Driven Traffic Simulation for an Intersection in a Metropolis

Chengbo Zang, Mehmet Kerem Turkcan, Gil Zussman, Javad Ghaderi, Zoran Kostic
Electrical Engineering, Columbia University
{cz2678, mkt2126, gil.zussman, jg3465, zk2172}@columbia.edu

## Abstract

*We present a novel data-driven simulation environment for modeling traffic in metropolitan street intersections. Using real-world tracking data collected over an extended period of time, we train trajectory forecasting models to learn agent interactions and environmental constraints that are difficult to capture conventionally. Trajectories of new agents are first coarsely generated by sampling from the spatial and temporal generative distributions, then refined using state-of-the-art trajectory forecasting models. The simulation can run either autonomously, or under explicit human control conditioned on the generative distributions. We present the experiments for a variety of model configurations. Under an iterative prediction scheme, the way-point-supervised TrajNet++ model obtained 0.36 Final Displacement Error (FDE) in 20 FPS on an NVIDIA A100 GPU.*

## 1. Introduction

Accurate modeling and reconstruction of traffic flows in simulation environments is important for solving transportation problems in modern cities [10]. Simulation of traffic trajectories within intersections of a metropolis involves consideration of realistic car movements, human decisions and interactions, environmental constraints, and various forms of social regulations.

Conventional simulation systems are often built bottom-up where the state space, rules of interactions, and policies are unambiguously defined beforehand. This can be challenging given the complex nature of real-world applications. Moreover, most existing simulation systems target traffic flow control and optimization, while lacking realistic fine-grained details of interactions between traffic participants. It is also quite challenging for such systems to model human decisions, where the behavior of each agent can be spontaneous, or affected by other agents as well as environmental constraints in the scene.

To address these challenges, this study uses a data-driven approach leveraging the data acquired from a real traffic intersection situated in a busy urban environment. We utilize

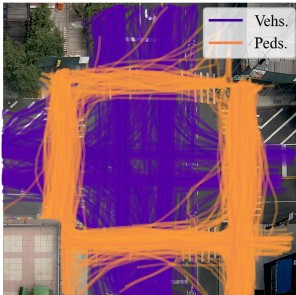
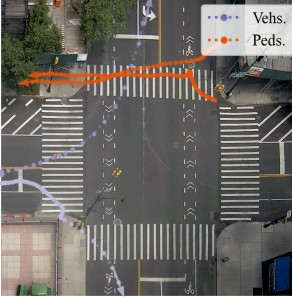
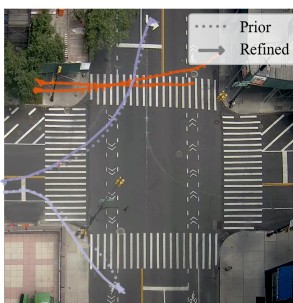

(a) Collected trajectories  (b) Categorized trajectories

(c) Coarse prior trajectories  (d) Refined trajectories

Figure 1. **Overall workflow of agent generation. (a)** Real-world trajectories collected from the intersection (vehicles in purple and pedestrians in orange). **(b)** Examples of different types of trajectories categorized by GMMs. Each color represents a different GMM component. **(c)** Coarse way-points sampled from GMMs and interpolated prior trajectories (denoted by dashed lines, where the circles are the sampled way-points). **(d)** Final trajectories refined by deep forecasting models (solid lines) compared to the coarsely sampled prior trajectories (dashed lines) in (c).

statistical priors and deep-learning-based trajectory forecasting models to capture the complex dynamics of traffic participants in real-world scenarios.

## 2. Related Work

**Traffic Simulation.** Conventional traffic modeling methods evolved largely from statistical physics [7]. These methods require heavy simplification assumptions and precise rule definitions. Modern approaches involve Deep Rein-

forcement Learning [21], evolutionary algorithms [24], or other state-space models [2]. However, these techniques often struggle in real-world scenarios due to the intractable size of states and policies. Most simulation systems focus on vehicle flows and exclude the role of pedestrians. More recent work includes the modeling of vehicle-pedestrian interaction such as Social Force Model [4], which has been adopted for the prediction of pedestrian motions [15, 33].

**Trajectory Forecasting.** Deep Neural Networks (DNNs) are used for predicting future motions of pedestrians and vehicles [28]. The key architectural component is often a sequential model (*e.g.* Recurrent Neural Network or Transformer) which autoregressively generates future predictions based on past observations [3, 12]. Some models take a generative approach and predict the embeddings of future trajectories from latent distributions to account for varying data patterns or noise using Generative Adversarial Networks (GANs) [14, 19], Generative Adversarial Imitation Learning (GAIL) [6, 8] or Conditional Variational Auto-Encoders (CVAEs) [5, 23, 30, 33]. Specially designed modules are introduced when modeling interactions in multi-agent scenarios by pooling [3, 13], attention operation [12, 29], or Graph Neural Networks [20, 23, 30]. Many architectures choose to incorporate auxiliary supervision using coarse way-points [9, 22] or final destinations [17, 31] of agent trajectories to boost model performances.

## 3. Method

### 3.1. Data Collection

We utilize a high-elevation camera overlooking a metropolis intersection. We fine-tuned a YOLOv8 object detection model [16] for pedestrians and vehicles, then collected real-world trajectory data under the tracking-by-detection paradigm featuring the BoT-SORT algorithm [1].

To underline the entry and exit locations for each agent for statistical analysis, we pre-processed the collected data by filtering out the trajectories that unexpectedly terminate in the middle of the intersection (due to occlusions or failure of the detection-tracking models). The filtered trajectories were then uniformly resampled to align at 30 FPS. Fig. 1a shows several processed trajectories overlaid on top of each other. Details about the dataset are described in Sec. 4.1.

### 3.2. Statistical Analysis

The distributions of pedestrian and vehicle trajectories exhibit clear dependencies both spatially and temporally (Fig. 1b and Fig. 2). It is intuitive to model them using conditional generative models, where the new agents would be generated by sampling from the distributions during simulation. At this stage of the study, we adopted Gaussian Mixture Model (GMM) for this purpose. We will explore

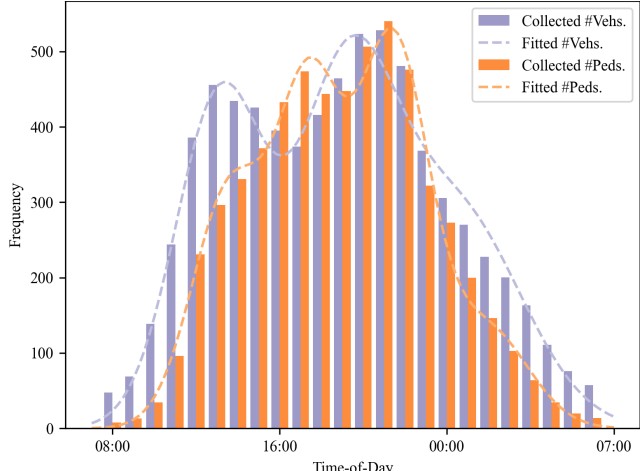

Figure 2. **Distribution of agent densities over** 24 **hours.** The bars show the collected number of agents while the dashed lines delineate the fitted pedestrian and vehicle frequencies, respectively. The $x$-axis is the ToD shifted to begin at 8:00 and end at 7:00 the next day, and the $y$-axis is the hourly average pedestrian and vehicle counts in the entire dataset.

models such as conditional GANs [25] or CVAEs [9] in future studies.

**Temporal Agent Density.** Fig. 2 gives the distribution of agent densities traveling through the intersection over different time-of-day (ToD). We assume that the ToD when agents enter the intersection is centered around a few peak hours (*e.g.* getting to work during the daylight or returning home at nighttime) and fit the mean pedestrian and vehicle densities using two GMMs (with 4 components for pedestrians and 3 components for vehicles, values determined by experiments). We denote their time-dependent distribution by

$$N_t \sim p_{tod}(N \mid t), \tag{1}$$

where $N_t$ is the total number of agents at time $t$.

**Spatial Trajectory Categorization.** In the case of urban intersections, the agent trajectories are generally more confined to follow specific patterns dictated by the layout of intersections, traffic rules, social regulations, and environmental constraints [11]. We propose to characterize the trajectory of each agent by: 1) the position and velocity at the point of entry into the intersection $\boldsymbol{x}(0), \boldsymbol{x}'(0) \in \mathbb{R}^2$; 2) the position and velocity at its exit from the intersection $\boldsymbol{x}(T), \boldsymbol{x}'(T) \in \mathbb{R}^2$; 3) the total time elapsed $T \in \mathbb{R}$ between its entry and exit; and 4) $|\mathcal{K}| = 20$ way-points sampled evenly along the trajectory $\boldsymbol{x}(\mathcal{K}) \in \mathbb{R}^{2|\mathcal{K}|}$, with sampling time $T/K$. Thus a vectorized representation of each agent can be given by $\boldsymbol{z} = \left[\boldsymbol{x}(0), \boldsymbol{x}'(0), \boldsymbol{x}(T), \boldsymbol{x}'(T), \boldsymbol{x}(\mathcal{K}), T\right] \in \mathbb{R}^{2|\mathcal{K}|+9}$. We model the distribution of different types of trajectories us-

ing a GMM with $M = 12$ components fitted respectively for pedestrians and vehicles, denoted as

$$\boldsymbol{z} \sim p_{gmm}(\boldsymbol{z}) = \sum_{m=1}^{M} w_m \, \mathcal{N}(\boldsymbol{z} \mid \boldsymbol{\mu}_m, \boldsymbol{\Sigma}_m), \qquad (2)$$

where $\mathcal{N}(\boldsymbol{\mu}, \boldsymbol{\Sigma})$ is a multivariate Gaussian and $w_m$ is the weight of component $m$. Examples of categorized trajectories from some GMM components are illustrated in Fig. 1b. Pedestrians and vehicles from six different components are plotted in different colors.

## 3.3. Generation of Prior Trajectories

---

**Algorithm 1** Prior Trajectory Generation Function.

---

**Input:** $N$                                     ▷ Number of agents
**Input:** $\mathcal{C}$                           ▷ Optional auxiliary conditions
**Output:** $\boldsymbol{x}_{pr}^{(1:N)}$          ▷ Generated prior trajectories

1: **function** PRIORGEN($N, \mathcal{C}$)
2:     **for** $i \leftarrow 1$ to $N$ **do**
3:         $\boldsymbol{z}_{\mathcal{C}}^{(i)} \sim p_{gmm}(\cdot \mid \mathcal{C})$      ▷ Agent sampling
4:         $\boldsymbol{x}_{pr}^{(i)} \leftarrow$ SPLINE($\boldsymbol{z}_{\mathcal{C}}^{(i)}$)      ▷ Resampling
5:     **end for**
6:     **return** $\boldsymbol{x}_{pr}^{(1:N)}$
7: **end function**

---

The algorithm for the generation of new agents during simulations is illustrated in Algorithm 1. We start by sampling pedestrians and vehicles from their corresponding GMMs. Auxiliary conditions can be provided to insert more control into the sampling process. For example, if one wishes to sample agents from specific GMM components (*i.e.* pedestrians or vehicles going in specific directions) in some set $\mathcal{C}$, then the GMM can be modified as

$$\boldsymbol{z}_{\mathcal{C}} \sim p_{gmm}(\boldsymbol{z} \mid \mathcal{C}) = \sum_{m \in \mathcal{C}} \hat{w}_m \, \mathcal{N}(\boldsymbol{z} \mid \boldsymbol{\mu}_m, \boldsymbol{\Sigma}_m), \qquad (3)$$

where $\hat{w}_m = w_m / \sum_{n \in \mathcal{C}} w_n$ is the adjusted component weight. This is exemplified in Sec. 4.2.

The sampled $\boldsymbol{z}$ can serve as good priors that provide high-level control over agent motions. However, this is not fine-grained enough due to the basic limitation that GMMs take no consideration of agent interactions or other environmental constraints. Sec. 3.4 describes a deep-learning-based refinement approach.

Note that the sampling times of the way-points $T^{(i)}/K$ are not uniform across different agents because their trajectories may have drastically different time elapsed $T^{(i)}$. Given that $\boldsymbol{z}$ also contains the position and velocity at both ends, we fit the trajectory with Cubic Splines [32], obtaining a piece-wise interpolating polynomial with time-continuous acceleration. We then evaluate the polynomial with a fixed

time interval $\Delta t = 0.4$s (2.5 FPS), obtaining a prior trajectory $\boldsymbol{x}_{pr}$ as inputs to deep-learning-based trajectory forecasting models.

Fig. 1c illustrates the prior trajectories of several pedestrians and vehicles. Their way-points are generated from the GMM components shown in Fig. 1b, followed by interpolation as described above.

## 3.4. Deep-Learning-Based Trajectory Refinement

To model agent interactions and other latent patterns in their motions, we adopt the TrajNet++ model [17], a DNN featuring an LSTM and a grid-based pooling module that deals with agent interactions. The model takes $L_{ob} = 8$ steps (3.2s) of past observations to predict $L_{pd} = 12$ steps (4.8s) into the future. The model operates in a goal-supervised manner, *i.e.* the agent positions at the end of the prediction window are also provided to the model as auxiliary inputs. The choice of $L_{ob}$ and $L_{pd}$ in our dataset follows from public benchmarks [18, 26, 27]. Sec. 5 presents more experiments comparing different $L_{pd}$-s.

At each time-step $t$, we combine $L_{ob}$ steps of previous trajectories from all agents in the scene as $\boldsymbol{x}_{ob} := \boldsymbol{x}(\boldsymbol{t}_{ob})$ (we use the sampled $\boldsymbol{x}_{pr}$ in case of newly generated agents with no past predictions), along with the temporal target locations (*i.e. goals*) of trajectories taken from $\boldsymbol{x}_{pr}$ at the end of the prediction window $\boldsymbol{x}_{tg} := \boldsymbol{x}_{pr}(t + L_{pd}\Delta t)$ as model inputs. The model then predicts

$$\boldsymbol{x}_{pd} := \boldsymbol{x}(\boldsymbol{t}_{pd}) = \text{DNN}\left(\boldsymbol{x}_{ob}, \boldsymbol{x}_{tg}\right), \qquad (4)$$

with

$$\begin{cases} \boldsymbol{t}_{ob} = [t - (L_{ob} - 1)\Delta t, \dots, t] \in \mathbb{R}^{L_{ob}} \\ \boldsymbol{t}_{pd} = [t + \Delta t, \dots, t + L_{pd}\Delta t] \in \mathbb{R}^{L_{pd}} \end{cases}. \qquad (5)$$

The model iteratively takes previous predictions as inputs while being supervised by temporal target locations taken from the priors. Fig. 1d shows the agent trajectories refined from Fig. 1c. We also explored several other model architectures and supervision schemes in Sec. 5.

## 3.5. Simulation Algorithm

The simulation algorithm is summarized in Algorithm 2. It maintains a set of active agents $\mathcal{A}_{ac}$. At each iteration, we **(i)** obtain the expected total number of agents $N_t$ from $p_{tod}$, **(ii)** generate prior trajectories of new agents $\boldsymbol{x}_{pr}$ from $p_{gmm}$ accordingly, and **(iii)** add them into $\mathcal{A}_{ac}$. Then the observations $\boldsymbol{x}_{ob}$ and target locations $\boldsymbol{x}_{tg}$ of all agents are sliced from $\mathcal{A}_{ac}$ (as in Eq. (4)) to construct DNN inputs and generate refined trajectories $\boldsymbol{x}_{pd}$, which are then concatenated to the historical data in $\mathcal{A}_{ac}$. Any agent whose current position $\boldsymbol{x}_t^{(i)} := \boldsymbol{x}^{(i)}(t)$ reaches its expected destination $\boldsymbol{x}_T^{(i)} := \boldsymbol{x}_{pr}^{(i)}(T^{(i)})$ will be considered to have exited the intersection and removed from $\mathcal{A}_{ac}$.

**Algorithm 2** Simulation Algorithm.

---

**Require:** $p_{tod}, p_{gmm}$  ▷ Distributions
**Require:** $\Delta t$  ▷ Simulation interval
1: $\mathcal{A}_{ac} \leftarrow \emptyset$  ▷ Set of active agents
2: **loop**
3:    $t \leftarrow t + \Delta t$  ▷ Generate new agents
4:    $N_t \sim p_{tod}(\cdot \mid t)$
5:    **if** $|\mathcal{A}_{ac}| < N_t$ **then**
6:        $N \leftarrow N_t - |\mathcal{A}_{ac}|$
7:        $\boldsymbol{x}_{pr} \leftarrow \text{PRIORGEN}(N, \mathcal{C})$
8:        $\mathcal{A}_{ac} \leftarrow \mathcal{A}_{ac} \cup \boldsymbol{x}_{pr}$
9:    **end if**
10:   $\boldsymbol{x}_{ob}, \boldsymbol{x}_{tg} \leftarrow \text{SLICE}(t, \mathcal{A}_{ac}, L_{ob}, L_{pd})$
11:   $\boldsymbol{x}_{pd} \leftarrow \text{DNN}(\boldsymbol{x}_{ob}, \boldsymbol{x}_{tg})$  ▷ DNN refinement
12:   **for** $\boldsymbol{x}^{(i)}$ in $\mathcal{A}_{ac}$ **do**
13:       $\boldsymbol{x}^{(i)} \leftarrow \text{CONCAT}(\boldsymbol{x}^{(i)}, \boldsymbol{x}_{pd}^{(i)})$
14:       **if** $\left\| \boldsymbol{x}_t^{(i)} - \boldsymbol{x}_T^{(i)} \right\| < \epsilon$ **then**  ▷ Check status
15:           $\mathcal{A}_{ac} \leftarrow \mathcal{A}_{ac} \setminus \boldsymbol{x}^{(i)}$
16:       **end if**
17:   **end for**
18: **end loop**

---

## 4. Experiments

### 4.1. Dataset and Evaluations

The collected data from the intersection were organized to fit different purposes. For object detection and tracking, 13k annotated images were collected sporadically over 5 years from a high-elevation camera overlooking the intersection. The fine-tuned YOLOv8 obtained 91.6 mAP for pedestrians and 98.7 mAP for vehicles, respectively.

For trajectory forecasting, tracked objects were collected for over 30 days containing time and bounding-box locations for 510k pedestrians and 250k vehicles. We uniformly sample 10k of 20-frame (8s) scenes and 10k of 40-frame (16s) scenes for trajectory forecasting model training and evaluation. Additionally, complete trajectories of 176k pedestrians and 215k vehicles were extracted from the collected data for the statistical analysis described in Sec. 3.2.

We trained trajectory forecasting models on the 20-frame scenes (with $L_{ob} = 8, L_{pd} = 12$) using smooth-L1 loss and adopted common performance metrics of Average-Displacement-Error (ADE, the RMSE between the predictions and ground-truths over all agents at all time-steps) and Final-Displacement-Error (FDE, the RMSE over all agents evaluated only at the last time-step of the prediction window) [17]. The models were trained on the standard scenes and evaluated in an iterative prediction scheme (described in Sec. 3.4) on the extended scenes to resemble the workflow during actual simulations. TrajNet++ with way-point supervision achieved the most desirable performance of 1.65

| Models | $L_{pd}$ | Goal | ADE / FDE (m) | FPS |
|---|---|---|---|---|
| LSTM | 12 | - | 2.34 / 4.25 | 288 |
| LSTM | 32 | - | 1.25 / 3.77 | 355 |
| Trajectron++ | 12 | - | 1.43 / 3.63 | 5 |
| Trajectron++ | 32 | - | 2.13 / 4.96 | 5 |
| TrajNet++ | 12 | Wpts. | 1.65 / **0.36** | 20 |
| TrajNet++ | 12 | Dest. | 1.92 / 0.51 | 20 |
| TrajNet++ | 32 | Dest. | **0.59** / 1.21 | 29 |

Table 1. Comparison of model performances on the 40-frame (16s) scenes on an NVIDIA A100. All models take $L_{ob} = 8$ frames (3.2s) of inputs. Some of them predict iteratively ($L_{pd} = 12$), others predict in one-shot ($L_{pd} = 32$).

ADE and 0.36 FDE, measured in meters. Experiments on different model architectures and configurations are provided in Sec. 5.

### 4.2. Controlled Simulation

The proposed simulation system can coarsely control agent trajectories with Eq. (3) in terms of where they enter and exit the intersection as well as a prior trajectory to follow. In Fig. 3a we purposefully sample south-bound pedestrians and left-turning vehicles whose prior trajectories meet at the middle of the crosswalk (the red ellipse), to see whether the trajectory forecasting model will correctly react to this situation.

As illustrated in Fig. 3b, the trajectory forecasting model forces both the vehicle and the crowd of pedestrians to slow down and deviate from their prior trajectories (denoted by the dashed red lines) to avoid a collision. This is a common practice that respects social norms and is expected to be observed in real-world scenarios.

### 4.3. Autonomous Simulation

Without inserting auxiliary conditions or other human control, the simulator is able to run autonomously and mimic different agent densities following Eq. (1) and spatial locations following Eq. (2).

## 5. Simulation Quality

### 5.1. Outliers

Adopting conditional generative models for trajectory categorization allows for the identification of outliers in the collected trajectories by calculating their likelihoods (Eq. (2)). In Fig. 3c, we show the outliers in pedestrians whose log-likelihoods are more than 20 times of standard deviations away from the dataset mean. Some of these outliers show a pedestrian making a turn-around; others show a pedestrian staying still in one location for an exceptionally long period of time.

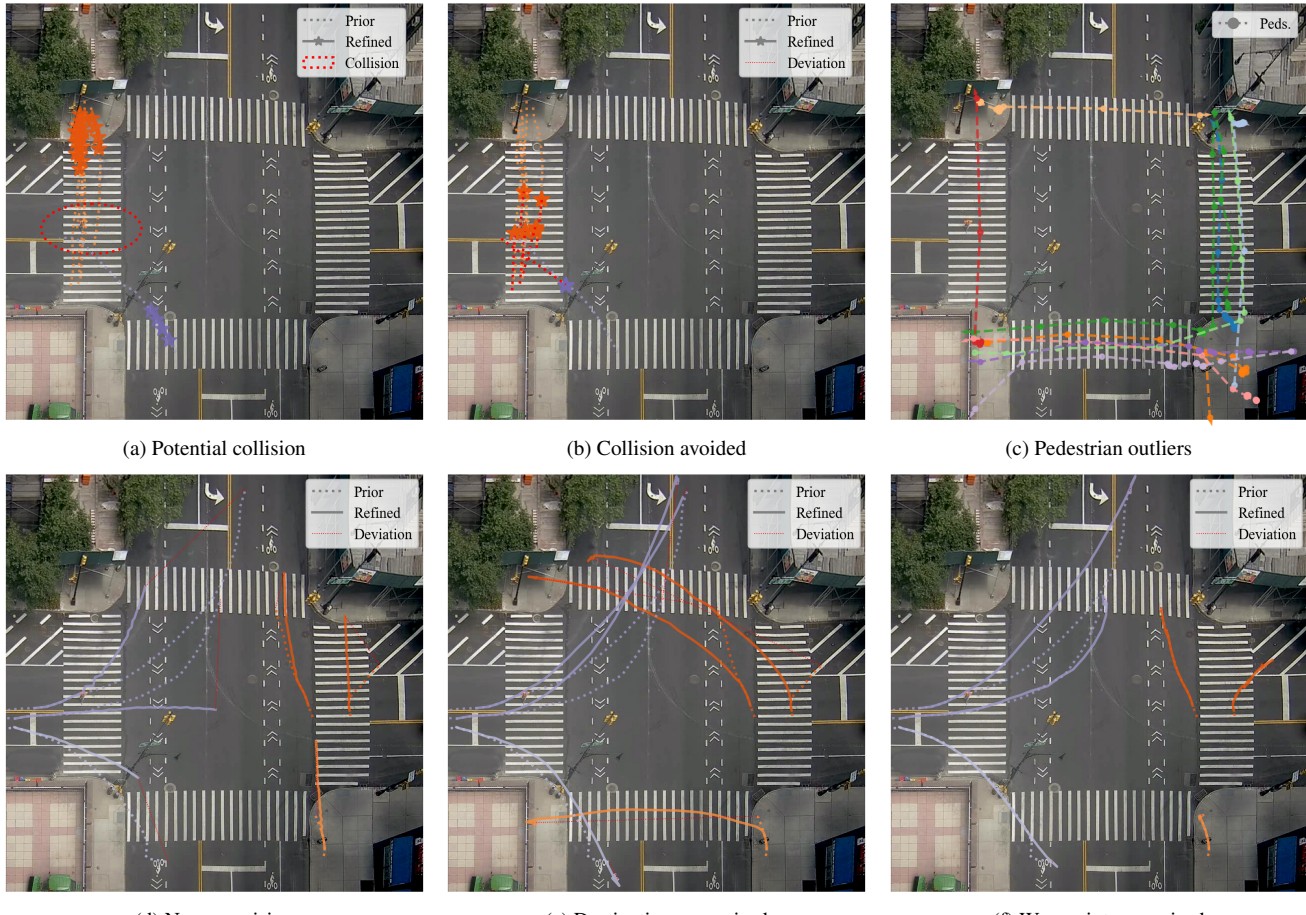

(a) Potential collision          (b) Collision avoided          (c) Pedestrian outliers

(d) No supervision          (e) Destination supervised          (f) Way-point supervised

Figure 3. **Simulation results and experiments. (a) - (b)** Controlled simulation of a potential collision and the reaction of the trajectory forecasting model. All agents slow down and deviate from their prior trajectories to avoid the collision. **(c)** Outliers identified in pedestrians by thresholding the likelihood of the trajectories. **(d) - (f)** Comparison of different supervision schemes for TrajNet++. Under the same priors, different results of refined trajectories are given by (d) no supervision, (e) final destination as supervision, and (f) way-points iteratively sampled from the priors as supervision.

## 5.2. Model Configuration

Beyond the standard metrics of ADE/FDE calculated over a predefined prediction window length $L_{pd}$, the measurement of simulation quality requires more careful considerations. Here we provide a brief discussion on system and model configurations based on current results.

**Goal Supervision.** We compared the refined trajectories from TrajNet++ before and after adding goal supervision (Fig. 3d *vs*. Figs. 3e and 3f). Significant improvements in refinement quality can be observed, where agents deviate less from their prior trajectories without external forces. Relevant results are also quantified in Tab. 1.

It is worth noting that the supervised models are often trained with fixed-length sequences and the agents are expected to reach their destinations in *exactly* $L_{pd}$ steps. This raises considerable issues in real-world deployment, since although it might not be difficult to know the destination of an agent (in our cases they are directly sampled), it can be challenging to know exactly when they will get there. Supervising the model with final destinations (Fig. 3e) resulted in notable overshoot in cases when an agent needs a longer time window to reach the destination than $L_{pd}$, while, in other cases, undershoot when they need a shorter window.

This can be mitigated by substituting the destinations with way-points taken from their prior trajectories at $L_{pd}$ steps ahead, with results shown in Fig. 3f. Alternating to smooth-L1 loss instead of MSE also dampens the overshoot. Agent motions tend to be more controllable by the statistical model (*i.e.* $p_{gmm}$), while being able to react and make deviations when necessary (Sec. 4.2).

**Choice of** $L_{pd}$**.** Using the 40-frame (16s) scenes, we further compared the performances of predicting iteratively by

training the model with $L_{pd} = 12$ (4.8s) and taking previous outputs as new inputs, *vs*. directly training the model to predict in one-shot with $L_{pd} = 32$ (12.8s). The results are given in Tab. 1. For TrajNet++, *Wpts.* denotes way-point supervision and *Dest.* destination supervision, as opposed to LSTM and Trajectron++ using no supervisions.

By comparison, the destination-supervised TrajNet++ model achieved the lowest ADE of $0.59$, while the way-point supervised version had a higher ADE but the lowest FDE of $0.36$. Higher FPS was generally obtained under larger $L_{pd}$ due to fewer operations beyond model inference (*e.g.* data preparation). Considering the aforementioned complexities of choosing appropriate $L_{pd}$ and applying destination supervision in practice, it is thus reasonable to use way-point supervision with shorter $L_{pd} = 12$ during actual applications, giving our chosen model for the experiments in Sec. 4.

## 6. Conclusion and Future Work

In this study, we propose a data-driven methodology for simulating the movement (trajectories) of agents within an intersection in a metropolis. We show that trajectory forecasting models are able to realistically govern agent motions under proper supervision by the statistical priors. The TrajNet++ model with way-point supervision was able strike a balance between the length of the prediction window and overall simulation quality by performing the predictions iteratively, achieving an FDE of $0.36$ under controlled experiments. However, we note that the presented models were trained and evaluated within a single traffic intersection, raising reasonable concerns on potential overfitting since traffic conditions may vary drastically across different locations. More comprehensive evaluation is needed to address the issue.

Future work will include **(a)** evaluation of alternative trajectory forecasting architectures and configurations, **(b)** incorporation of a larger number of intersections and more diverse traffic scenarios for better generalization, **(c)** exploration of other potential cases of agent interactions under controlled simulation, and **(d)** investigations on how to connect broader aspects of applications (*e.g.* collision alert, traffic light control, and more efficient deployment). We intend to incorporate the model with graphics engines where we can reconstruct the traffic scenarios of the intersection in the digital world.

## Acknowledgements

This work was supported in part by NSF grant CNS-1827923 and EEC-2133516, NSF grant CNS-2038984 and corresponding support from the Federal Highway Administration (FHA), NSF grant CNS-2148128 and by funds from federal agency and industry partners as specified in the Resilient & Intelligent NextG Systems (RINGS) program, and ARO grant W911NF2210031.

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
