# OpenReview forum: "Data-Driven Traffic Simulation for an Intersection in a Metropolis"
_thecvf.com/CVPR/2024/Workshop/POETS — CVPR 2024 Workshop POETS Oral_

### Official Review · Reviewer_8HEz · 2024-05-08

**Rating:** 7
**Confidence:** 3

**Review:**

In this paper, the authors propose a pipeline for modeling and simulating traffic at street intersections. The trajectory models are trained using real-world tracking data. When simulating an agent, a coarse trajectory is first sampled from a GMM and then refined by an LSTM.

Strengths:

The authors expend significant effort in preparing real-world data for the data-driven method, including five years of data for object detection and tracking, and 30 days for trajectory forecasting.

The GMM + LSTM model is sensible, where the GMM can generate diverse coarse trajectories, while the LSTM is capable of modeling nonlinear interactions.

Based on the existing data and the methodological backbone, several interesting follow-up works are proposed in the future work section, such as collision avoidance and traffic light control.

Weaknesses:

The diversity of the testing scenarios could be improved. Currently, only one intersection is showcased, while traffic patterns could vary significantly in different places.

The presentation of this work could be enhanced if the authors provided some videos.

Summary:

This paper presents a data-driven traffic modeling and simulation pipeline based on their collected data. The entire pipeline appears reasonable with promising future work. The project could be made more comprehensive with a better variety of tasks and data sources.

---

### Official Review · Reviewer_jdMS · 2024-05-13
**Review of submission 4**

**Rating:** 7
**Confidence:** 4

**Review:**

**Strengths**

- Novel Data-Driven Approach: The paper introduces a novel approach to simulate traffic at urban intersections using data-driven methods. The use of real-world data to train trajectory forecasting models is commendable, as it aims to reflect real-world complexities in traffic dynamics.
- Comprehensive Methodology: The methodology is thorough, encompassing data collection, statistical analysis, generation of prior trajectories, and trajectory refinement. The integration of Gaussian Mixture Models (GMMs) and deep-learning models like TrajNet++ for refining trajectories enhances the simulation's realism.
- Clear Visualization and Documentation: The paper includes detailed figures that illustrate the workflow and results effectively, enhancing understanding of the complex processes involved.
- Important potential use: the discussion of this work's future work of incorporating the model with graphics engines is a very important and valuable direction.

**Weaknesses**

- Topic of The Workshop. This work discusses the simulation of humans and traffic. However, the discussion of the role of humans in the traffic is missing. Will humans be important in this research direction? How will humans affect the traffic (an interesting multi-agents question)?
- Model Complexity and Computation Cost: The reliance on sophisticated models and high-end GPUs (like NVIDIA A100) might limit the accessibility of the proposed system for users with fewer computational resources.
- Generalization Concerns: While the simulation provides good results in a controlled environment, the paper does not thoroughly discuss how the system performs across various types of intersections or under different traffic conditions.
- Potential for Overfitting: Given the detailed customization of the models to specific datasets, there is a risk of overfitting. More robust validation using diverse external datasets could strengthen the findings.

---

### Official Review · Reviewer_P3P4 · 2024-05-13
**The paper provides a data-driven traffic simulation by a series standard operations and provide insights for future work.**

**Rating:** 8
**Confidence:** 4

**Review:**

Overall, the paper has good illustration and interpretation, the project has practical relevance, and an innovative approach is designed to model complex urban traffic scenarios, making it a positive candidate for acceptance.

### Quality
1. The paper demonstrates a comprehensive methodology for traffic simulation in urban intersections. It utilizes real-world trajectory data for training trajectory forecasting models, and the method and training process are well-constructed with clear steps in data collection, processing, and model application.

2. The paper is also well organized with a clear interpretation of the result and case analysis. It is a good paper for reading.

### Clarity

1. The paper's figures and tables effectively support the textual content, providing clear visual summaries of the methodology and results.
2. However, some sections could benefit from more detailed explanations to avoid ambiguity: such as in table 1. what does the `Wpts` mean?
3. Some relevant work is suggested to include to provide a more comprehensive survey, such as for trajectory forecasting, another branch of method named Generative Adversarial Imitation Learning (GAIL):
[1] Choi S, Kim J, Yeo H. TrajGAIL: Generating urban vehicle trajectories using generative adversarial imitation learning[J]. Transportation Research Part C: Emerging Technologies, 2021, 128: 103091.
[2] Da L, Wei H. CrowdGAIL: A spatiotemporal aware method for agent navigation[J]. Electronic Research Archive, 2022, 31(2).


### Originality
The paper is of good originality.

### Significance

The significance of this work lies in its potential impact on urban planning, traffic management, and autonomous vehicle systems:
1. It helps in providing a more accurate and realistic simulation of traffic dynamics at intersections,
2. This research could help in designing better traffic control systems
3. This paper may have a potential influence in improving the safety and efficiency of urban transportation.

---

### Meta-Review · Program_Chairs · 2024-05-14

**Recommendation:** Accept (Oral)
**Confidence:** 5

**Metareview:**

The reviews collectively appreciate the paper's innovative, data-driven approach to modeling traffic dynamics at urban intersections using real-world data. Strengths include the detailed methodology, practical relevance, and clear visual documentation. Weaknesses involve potential overfitting, the need for broader testing scenarios, and generalization concerns. Also, the discussion of the role of virtual humans in the traffic is asked to better match this workshop.
Overall, the paper is recommended for acceptance due to its significant potential impact on urban planning and traffic management, with suggestions for clearer explanations and an expanded literature review.

---

### Decision · Program_Chairs · 2024-05-14

Accept (Oral)